# Non-universal current flow near the metal-insulator transition in an oxide interface

Eylon Persky[1], Naor Vardi[1], Ana Mafalda R. V. L. Monteiro[2], Thierry C. van Thiel[2], Hyeok Yoon[3,4],
Yanwu Xie [3,4,5], Benoît Fauqué[6], Andrea D. Caviglia [2], Harold Y. Hwang [3,4], Kamran Behnia [7],
Jonathan Ruhman[1] & Beena Kalisky [1✉]

In systems near phase transitions, macroscopic properties often follow algebraic scaling laws, determined by the dimensionality and the underlying symmetries of the system. The emergence of such universal scaling implies that microscopic details are irrelevant. Here, we locally investigate the scaling properties of the metal-insulator transition at the $LaAlO_3/SrTiO_3$ interface. We show that, by changing the dimensionality and the symmetries of the electronic system, coupling between structural and electronic properties prevents the universal behavior near the transition. By imaging the current flow in the system, we reveal that structural domain boundaries modify the filamentary flow close to the transition point, preventing a fractal with the expected universal dimension from forming.

[1] Department of Physics and Institute of Nanotechnology and Advanced Materials, Bar-Ilan University, Ramat Gan, Israel. [2] Kavli Institute of Nanoscience, Delft University of Technology, Delft, The Netherlands. [3] Geballe Laboratory for Advanced Materials, Department of Applied Physics, Stanford University, Stanford, CA, USA. [4] Stanford Institute for Materials and Energy Sciences, SLAC National Accelerator Laboratory, Menlo Park, CA, USA. [5] Department of Physics, Zhejiang University, Hangzhou, China. [6] JEIP, USR 3573 CNRS, Collège de France, PSL Research University, Paris, France. [7] Laboratoire Physique et Etude de Matériaux (CNRS-Sorbonne Universitè), ESPCI Paris, PSL Research University, Paris, France. ✉email: beena@biu.ac.il

Universal scaling laws in systems near phase transitions are one of the hallmark discoveries of twentieth century physics; near critical points, the thermodynamic properties of fundamentally different systems follow the same algebraic scaling laws[1]. This property allows us to strip complex systems of their microscopic details and characterize them using only their dimensionality and underlying symmetries. For example, at a critical porosity, water percolating through the ground flows along a subdimensional fractal, whose dimension is independent of the soil's details[2]. Fractals with the same universal dimension emerge in a variety of other systems, such as forest fires, galactic structures, and metal-insulator transitions[3].

Metal-insulator transitions in complex materials, like transition metal oxides, offer an experimental platform for testing the applicability of the universal scaling description. There are clear experimental observations of critical scaling consistent with percolation in a variety of transition metal oxides[4,5]. However, the electronic transition is often intertwined with other material properties, such as magnetic and structural orders[6,7]. Although such couplings can change the dimensionality or symmetries of the electronic systems, it is unclear whether these orders interfere with the expected universal criticality[8–11]. Thus, to determine the mechanism driving the electronic transition, it is essential to resolve the interplay between the electronic and structural orders.

Investigating how domain patterns modify the electronic transition requires local tools, capable of both resolving the patterns, and discerning their effect on electronic properties. We therefore used scanning superconducting quantum interference device (SQUID) microscopy to image the current flow in proximity to the gate-tunable metal-insulator transition (MIT) at the LaAlO$_3$/SrTiO$_3$ (LAO/STO) interface. On the metallic side, the conductivity of LAO/STO is weakly modulated (in space) over structural domain patterns, which emerge from the STO substrate[12–15]. They lead to narrow, elongated, highly conducting channels, with lengths comparable to those of mesoscopic devices[16]. In LAO/STO, there is a gate tunable superconductor-insulator transition[17–19]. At higher temperatures, an MIT has been observed through resistivity[20–22] and compressibility measurements[23]. Resistivity measurements revealed a percolation type transition[19–21], but sample-to-sample variations near the critical point have been reported[20,24].

Here, we reveal that in the presence of domain boundaries, the critical behavior of LAO/STO is not universal. We show that metallicity persists along domain boundaries, as the bulk 2D system turns insulating. As a result, the current carrying backbone cannot scale with the expected universal fractal dimension. Using random resistor network simulations, we show that the lack of a universal backbone coexists with universal scaling of the conductivity, and that the conductivity threshold is size dependent. This combination of universal and non-universal properties suggests that transitions in complex materials must be probed over multiple length scales, to discern their true properties.

## Results

To map the current density, we imaged the magnetic fields generated by the current, and used Fourier analysis to reconstruct the current density ("Methods" and Supplementary Note 2). We studied metallic interfaces, with LAO thicknesses of 10 u.c. and 12 u.c. In such samples, a metal-insulator transition can be achieved by tuning the metallicity using electrostatic gating. We used a back-gate geometry, where a gate voltage ($V_G$) is applied between the 2D interface, and a metallic surface at the bottom of the sample. Reducing $V_G$ pushed the sample toward its insulating phase. We first show that, in the absence of domain boundaries, the transition is locally consistent with percolation. In a mono-

domain, on the metallic side, the current density was overall homogeneous, except for one local reduction (Fig. 1a). In the metallic phase, such local region with reduced current density could reflect a local reduction in the conductivity (roughly 50% of the neighboring area), possibly due to defects. Disorder increased as we lowered $V_G$: more low current density patches appeared, and existing patches expanded (Fig. 1b, c). Particularly, previously disjoint patches merged to form large regions of low current density, distorting the current flow. The gradual increase in inhomogeneity is consistent with previous transport studies[19–21], which suggest the MIT in LAO/STO occurs via percolation. There are two spatially separated phases – metallic and insulating, and the transition occurs when the metallic phase percolates through the insulating one.

Next, we show that domain boundaries prevent the current carrying backbone from forming a fractal. Figure 2a shows the current flow in a mesoscopic device, with domain boundaries oriented perpendicular to the overall direction of the current flow. Current flow along channels with such orientation is unfavorable, and therefore, the amount of current found on these channels measures the level of inhomogeneity in the surrounding mono-domain regions (Supplementary Note 9). Indeed, as we reduced $V_G$, the current along the boundaries increased together with the disorder in the mono-domains. At the lowest $V_G$, the current focused along complicated, curved paths in the mono-domain regions, interrupted by straight lines along the boundaries. Although the distorted current flow suggests a large inhomogeneity in the conductivity landscape, areas of low current density are not necessarily entirely insulating. There are several conductivity landscapes that can produce such current flow. One possibility is that these areas contain metallic regions disconnected from the current-carrying backbone. Another option is "dead end" conducting paths. Such paths originate from the backbone, but terminate at insulating regions. For such "dangling bonds", we can estimate from the data the conductivity reduction at the insulating regions. The contrast of current densities on and off the backbone suggests that their conductivity is at most 1-5% of the conductivity of the backbone.

The current flow along the boundaries is inconsistent with the expected fractal structure: fractals scale sub-dimensionally with the system size ($d_f = 1.4$ for 2D percolation)[3], which is inconsistent with current paths that are one dimensional over a finite region. Thus, in a multi-domain system, the current carrying backbone cannot scale with the correct universal exponent.

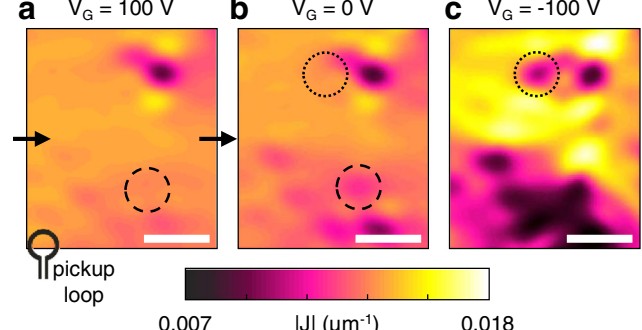

**Fig. 1 Disorder and percolation in a mono-domain.** Current density (magnitude, |J|) images of a mono-domain region (sample H1), at various gate voltages, $V_G$: 100 V (**a**), 0 V (**b**), and −100 V (**c**). Reducing $V_G$ increases the inhomogeneity of the current flow, as additional patches of low current density appear (circles), and isolated patches cluster together (**c**). The arrows in panel a mark the overall direction of the current flow. The SQUID's pickup loop, (diameter 1 μm), is not drawn to scale. Scale bars, 15 μm.

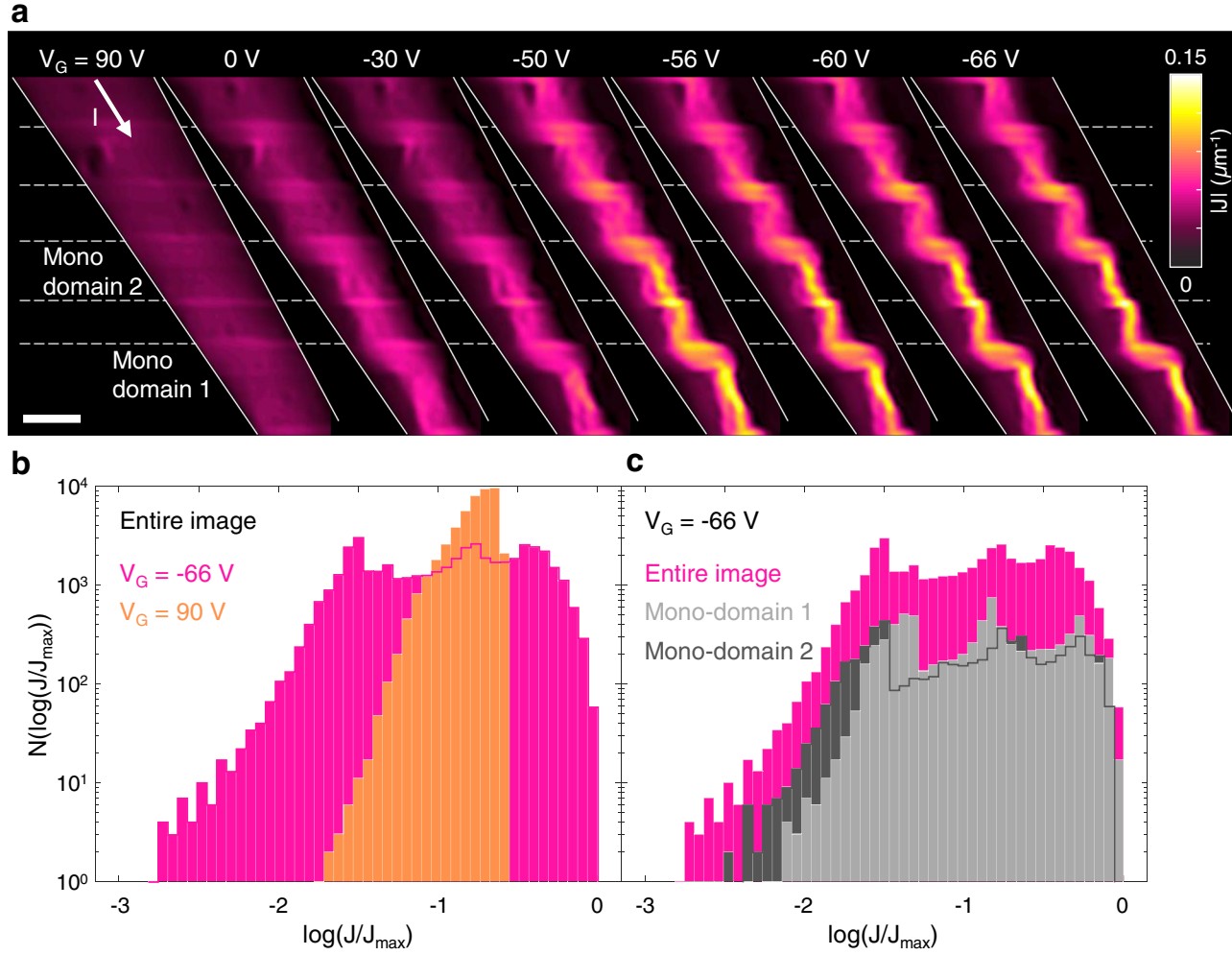

**Fig. 2 Non-universal current backbone in a multi-domain device. a** Current density images of a multi-domain region of sample C1, as a function of $V_G$. Reducing $V_G$, mono-domain regions became more inhomogeneous, while the boundaries remained conducting. As a result, more current flowed along the boundaries. The rightmost image shows complicated current paths in the mono-domain regions, interrupted by the boundaries. In such patterns the current-carrying backbone cannot have the universal fractal scaling dimension. The solid white lines indicate the edges of the pattern. The dashed white lines indicate positions of domains. Scale bar, 30 μm. **b** Logarithmically binned histogram of the current density of the leftmost and rightmost maps in **a**, transitioning from highly homogeneous conductivity on the metallic side, characterized by a narrow current distribution to disordered flow closer to the MIT, where the histograms spans three decades of current density. **c** Histograms of two mono-domain regions at $V_G = -66$ V, showing variations in the current distributions between different mono-domains, particularly in the width of the distributions, and in the low-current behavior.

Next, we consider the statistical distribution of the current density. Figure 2b shows logarithmically binned histograms of the current densities, in the metallic phase ($V_G = 90$ V) and close to the transition ($V_G = -66$ V). The histograms transition from a narrow distribution, consistent with homogeneous conductivity in the metallic phase, to a wide distribution at $V_G = -66$ V, which spans three decades of current density. We note that the current carrying backbone we observe is resolution limited, and may contain a finer structure. In this case, the maximal current density could be larger than the value extracted from the data, thus extending the low current tail of the histograms. We also considered the current distributions in different mono-domains, compared with that of the overall image. At $V_G = -66$ V, histograms of different mono-domains vary with respect to the histogram of the overall image (Fig. 2c). Particularly, the overall distribution contains a low-current tail extending one decade more than that of mono-domain 1. Because the metallic boundaries separate the system into smaller, independent regions, the variations in the histograms may be attributed to a varying disorder landscape in the system. The relevant length-scale for

percolation in the mono-domains is set by the distance between domain boundaries, not the overall system size. Thus, the critical behavior is detail dependent, rather than universal.

Even though the current-carrying backbone we find here does not have the universal form expected from a percolation transition, previous resistivity measurements yielded the expected universal critical exponents[19–21]. Compared to semiconductor based 2DEGs[25], the MIT in LAO/STO is anomalous. First, the insulating phase in LAO/STO has a weak temperature dependence[17,20,21], and a crossover between $dR/dT > 0$ and $dR/dT < 0$ is often difficult to observe. Instead, the MIT is identified by a sharp rise in resistivity over a narrow range of gate voltages. The insulating phase is often accompanied by onset of non-linear IV characteristics[20]. We studied the transport properties of two LAO/STO samples, and observed non-linear IV curves and large drop in conductivity below a critical $V_G$ (Supplementary Note 3), confirming our samples have bulk characteristics of a gate tunable MIT. To see how a non-universal backbone can coexist with universal scaling of the resistivity, we used random resistor network (RRN) simulations. The

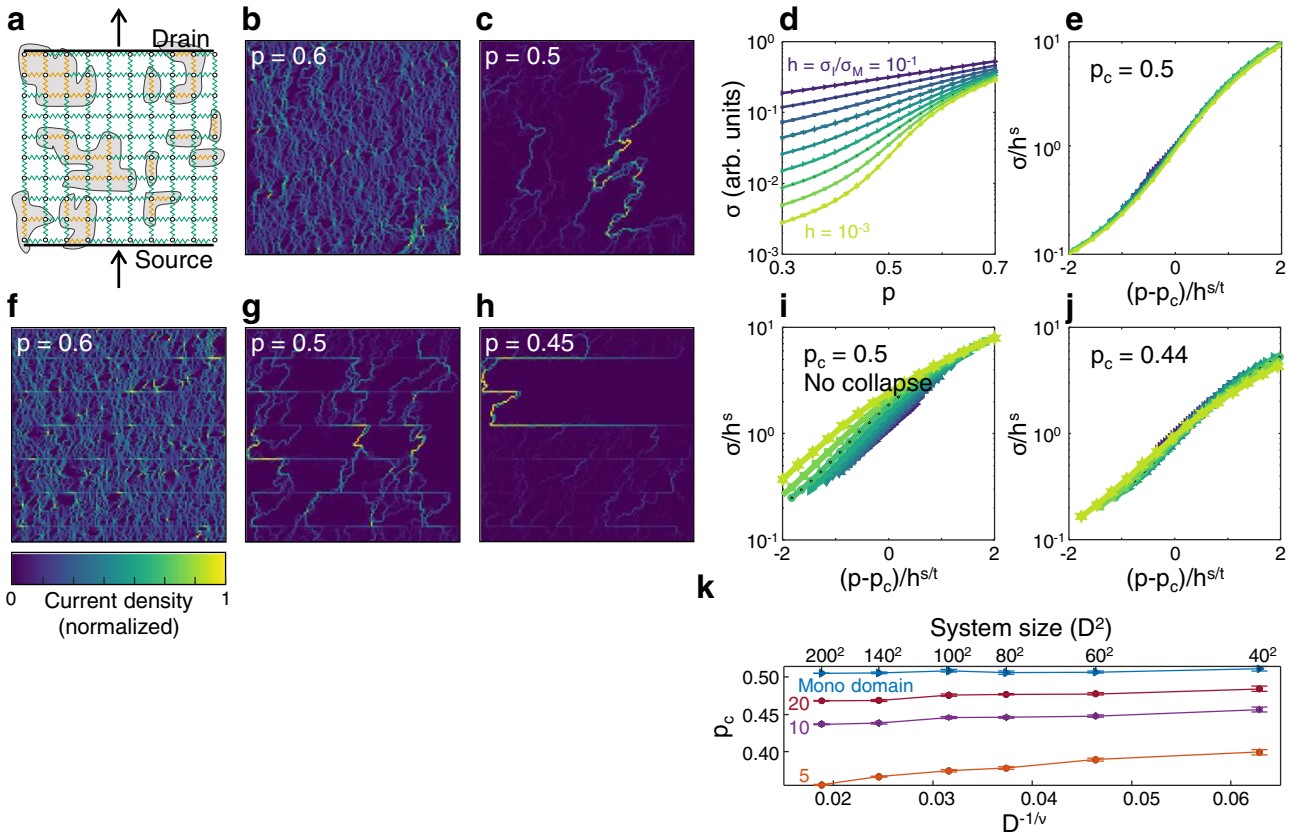

**Fig. 3 Size dependence of the metal-insulator transition. a** Illustration of the random resistor network, showing isolated insulating patches (yellow nodes) within the metallic (green nodes) network. **b**, **c** Current density maps obtained from individual realizations of a 128*128 network, at $p = 0.6$ (**b**) and $p = 0.5$ (**c**). The current was sourced at the bottom side of the square, and drained at the top. At $p = 0.5$, the current flows along the expected subdimensional fractal. **d** The conductivity of the network, plotted as a function of $p$, for various bond conductivity ratios $h$. **e** The data in d rescaled with the universal critical exponents $s = 0.5$ and $t = 1.3$, showing data collapse at the critical concentration, $p_c = 0.5$. Representative realizations of the model with domain boundaries, for $p = 0.6$ (**f**), $p = 0.5$ (**g**), and $p = 0.45$ (**h**). The current carrying backbone in g no longer consists of a single complicated path. Instead, shorter complex paths appear at a lower concentration (**h**), interconnecting the domain boundaries. The boundaries introduce long, straight lines into the backbone, inconsistent with the expected size scaling of a fractal. **i**, **j** Rescaled conductivity data for the critical concentration of the standard model, $p_c = 0.5$ (**i**), and for the shifted critical point $p_c = 0.44$ (**j**). **k** Size dependence of the critical concentration, for networks with various domain wall spacings. The mono-domain network shows no size dependence, while the critical concentrations of multi-domain RRNs is lower for larger systems sizes. The size and detail dependence suggest that the transition is not universal. The error bars represent two standard deviations from the mean fitted critical point.

simulations enable us to study how the critical behavior depends on the system size, a parameter not easily tuned in experiments. We show that, although conductivity can be rescaled with the correct critical exponents, the critical concentration strongly depends on the exact domain pattern, and the overall sample size. As a result, the transition is suppressed for macroscopic systems, and the critical scaling only occurs as a finite-size effect.

We begin by considering the scaling properties of a mono-domain. To model mono-domains, we used a standard network on a 2D square lattice (Fig. 3a, "Methods"). Each node in the network is either metallic (with probability $p$) or insulating (with probability $1 - p$). Although the RRN treatment simplifies the transport properties of LAO/STO, it enables us to study the universal scaling behavior near the critical point; similar models were previously used to describe the superconductor-metal-insulator transition in LAO/STO mono-domains[26,27]. This mono-domain network has a phase transition when the concentration of metallic nodes, $p$, is $p = p_c = 0.5$[3,28]. At $p_c$, current only flows through a small portion of the metallic bonds, forming the expected subdimensional fractal. Correspondingly, the global conductivity of the network shows clear data collapse (Fig. 3d, e)

when the data are rescaled according to[28]

$$\frac{\sigma}{h^s} = \varphi\left(\frac{p - p_c}{h^{t/s}}\right), \tag{1}$$

where $h = \sigma_I/\sigma_M$ is ratio between the conductivities of metallic and insulating bonds, $\varphi$ is a universal scaling function, and $s = 0.5$ and $t = 1.3$ are the universal scaling exponents for percolation on a 2D square lattice.

We included domain boundaries by modifying the RRN. We considered a set of boundaries oriented perpendicular to the overall direction of current flow, to mimic the experimental data (Fig. 2d–g). We modeled the boundary as a set of highly conductive nodes, with conductivity $\sigma_B > \sigma_M$. In this model, the metallic fraction, $p$, is a parameterization of $V_G$. Lower $p$ corresponds to more negative $V_G$. We simplified the transport on the highly conducting channels by assuming it is independent of the gate voltage (metallic fraction), and that the domain structure does not change with the applied gate. Thus, other than $p$, the model parameters do not depend on the gate. We neglected gate induced domain motion because we did not observe it in the samples studied. Domain boundaries in LAO/STO are expected

to host 1D wave guides with enhanced carrier density[29,30]. We expect the detailed electronic properties to change the conductivity of the insulating phase. However, the current flow images show that the channels remain highly conductive even as the bulk turns insulating (Fig. 2a). It is therefore appropriate to neglect the detailed electronic structure when considering the effects on critical behavior. Comparing the current density maps at the presumed critical point, $p = 0.5$, confirms the experimental observation, that the current carrying backbone contains straight narrow lines, inconsistent with a fractal structure. As a result, the conductivity data no longer collapse when rescaled with $p_c = 0.5$ (Fig. 3i). The data collapse is recovered at a lower concentration (Fig. 3j). Thus, the introduction of domain boundaries pushes the critical point to a lower value. Recovering the universal scaling is possible because the overall conductivity of the system is determined by the current flow in the mono-domains, rather than the boundaries. Thus, measuring the conductivity probes the mono-domains, which do abide by the expected scaling laws.

Next, we show that the critical concentration depends on the system size, suggesting that the boundaries suppress the transition in the thermodynamic (large system) limit. We simulated additional RRNs, with various system sizes and domain patterns (Fig. 3k, Supplementary Note 7). Domain patterns in mesoscopic LAO/STO devices, often consist of long boundaries, arranged in comb-like patterns[16]. Thus, to describe large systems, we considered uniformly spaced boundaries, whose length is the system size. For each network, we found the critical concentration which recovered the scaling behavior. The resulting critical concentrations (Fig. 3k, Supplementary Note 7) depend on the orientation and spacing of the domain boundaries, and, crucially, on the overall network size. As the network size increases, the critical point is pushed to lower values, suggesting that there is no transition for large networks.

To understand the size dependent shift, we note that, in a standard RRN, the probability to percolate from one side of the sample to the other is given by $P \sim De^{-D/\xi}$, where $D$ is the lateral size of the sample and $\xi \sim |p - p_c|^{-\nu}$ is the correlation length (for 2D percolation, $\nu = 4/3$). In the thermodynamic limit ($D \to \infty$), this probability vanishes, and the sample becomes insulating. When domain walls are present, the largest distance between walls, $l_{max}$, controls the percolation probability, $P \sim De^{-l_{max}/\xi}$, which is no longer exponentially small in $D$. As a result, the critical probability depends logarithmically on $D$ (Supplementary Note 8), leading to the breakdown of the universal scaling form of Equation 1. Thus, we conclude that the transition in mesoscopic systems containing domain boundaries cannot be described in terms of finite-size scaling.

## Discussion

The simulations and experimental data therefore reveal a surprising combination of critical behaviors. In finite samples, local features (such as the current carrying backbone) do not show the expected universal structure, whereas global features such as the conductivity scale correctly, at a size-dependent critical concentration, because the conductivity is determined by the largest mono-domain region. These results suggest that sample-to-sample variations observed in LAO/STO samples at low carrier densities (see Supplementary Note 10 for discussion) can be explained by variations in the domain patterns. Thus, studying the critical properties of mesoscopic systems requires probing them both on local and global scales.

The lack of a fractal backbone also offers an opportunity to use the gate tunable MIT to study the electronic properties of individual domain boundaries. For example, orienting boundaries along the overall current flow could generate a 2D to 1D

crossover, as the current will flow entirely along the boundaries. In STO, this is a particularly promising application, as domain boundaries support a variety of properties, not available in the bulk: they serve as 1D ballistic wave guides[30–32] and host 1D superconductivity[29], electric polarity[13], and magnetic properties[33]. While the realization of such 1D devices requires better control over the size and orientations of domain boundaries, our results show how to decouple the 1D nanowire from the host 2D system.

To conclude, we have shown that, near the MIT in LAO/STO, domain boundaries prevent onset of universal critical behavior. The boundaries hinder the formation of a universal fractal current carrying backbone, and modify the scaling behavior of the conductivity, by making the critical point size dependent. Our results suggest that when electronic properties are coupled to other orders, critical behavior depends on the finite-sized domain patterns, making it non-universal.

Coupling between electronic and lattice structures is characteristic of transition metal oxides[6,7]; MITs often occur together with a magnetic or structural phase transition[8,11,34–39]. To understand the electronic transitions, it is essential to determine how the structural/magnetic transitions contribute, and whether they are the driving force behind the MIT. These questions remain under debate[36,37,40], because the structural and electronic properties are difficult to untangle. Our results suggest that, by modifying the filamentary flow, coupling between structural and electronic properties can create novel critical behavior. For example, near a structural transition, a metallic filament consisting of domain boundaries should be extremely susceptible to changes and can therefore strongly perturb the electronic transition.

Our results also offer a generic platform for engineering the current flow in other oxide heterostructures. Through interface engineering, the domain patterns of the substrate can be imprinted onto the structure of thin films. Coupling between carriers in thin films and the structural properties or phonon modes of their substrates have been reported in a variety of heterostructures[41–44]. Thus, engineering these couplings can generally lead to nanoscale control over the electronic properties and critical behavior of the films. This type of engineering could be a powerful tool for both fundamental and applied research. To achieve this, further work is required to improve the control over formation of domain patterns. Control over domain patterns (domain engineering) already has promising applications with ferroelectrics and ferromagnets[45], and techniques established for ferroics could be applied to heterostructures.

Finally, we come back to the relationship between algebraic scaling and fractals. We showed that scaling exponents of global properties do not determine the fractal dimension on the microscopic scale. Our results therefore provide a counter-example to the hyper-scaling relations, suggesting that algebraic scaling of macroscopic quantities does not always imply self-similarity on the microscopic scale.

## Methods

LAO films were deposited on TiO$_2$ terminated STO substrates, using pulsed laser deposition[46]. Growth conditions are described in Supplementary Table 1. The samples were patterned using an AlO$_x$ hard mask[47]. Electrical contacts to the sample were made by ultrasonic wire bonding directly to the interface. Alternating currents (1 μA–60 μA rms, with frequencies between 200 Hz and 2 kHz) were applied to the sample. The resulting magnetic flux was recorded with a scanning SQUID with a 1 μm pickup coil, using a lock-in amplifier. The current density was reconstructed using Fourier analysis[48], and representative raw magnetic flux data are shown in the Supplementary Fig. S1. Data shown in Fig. 1 were taken at 4.2 K and data shown in Fig. 2 were taken at 5.5 K. The gate leakage current was lower than 1 nA in all measurements.

RRN simulations were performed on square lattice grids with dimensions D × D. Each node was assigned one of three labels: metal, insulator, or domain boundary. Metal and insulator labels were assigned randomly. Metallic nodes were given

conductivity $\sigma_M = 1$, domain walls were given conductivity $\sigma_B = 5$, and the conductivity of insulating nodes was varied between simulations. Kirchhoff's rules were then applied for each site, to obtain a set of linear equations for the voltage drop on each resistor. The boundary conditions were set so that current is sourced on one side of the square and drained on the opposite site. The equations were solved via matrix inversion, after one equation had been eliminated, fixing the overall potential shift. The current on each node was then inferred using Ohm's law. The critical point was obtained by minimizing the distance between the rescaled curves (Eq. (1)), using $p_c$ as the only fitting parameter. Concentrations with $|p - p_c| < D^{-1/\nu}$ were not used for the fit, to avoid effects stemming from the finite dimensions of the network. Error bars were obtained by statistical bootstrapping.

## Data availability
The data that support the findings of this study are available from the corresponding author upon reasonable request.

## Code availability
The algorithms and simulation codes used in this study are available from the corresponding author upon reasonable request.

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

## Acknowledgements
We thank Herb A. Fertig, Ganapathy Muthry, Efrat Shimshoni and Brian Skinner for fruitful discussions. E.P., N.V., and B.K. were supported by European Research Council Grant No. ERC-2019-COG-866236, and Israeli Science Foundation grant no. ISF-1281/17. J.R. was supported by Israeli Science Foundation grant no. 967/19. A.D.C. was supported by European Research Council Grant No. ERC-2015-STG-677458, and by The Netherlands Organisation for Scientific Research (NWO/OCW) as part of the VIDI programme. B.K., and A.D.C. were supported by the QuantERA ERA-NET Cofund in Quantum Technologies (Project No. 731473). B.F. acknowledges the support from Jeunes Equipes de l'Institut de Physique du Collège de France and by a grant attributed by the Ile de France regional council. Work at Stanford was supported by the Department of Energy, Office of Basic Energy Sciences, Division of Materials Sciences and Engineering, under contract No. DE-AC02-76SF00515.

## Author contributions

E.P. and B.K. conceived the work and designed the experiments. E.P, N.V., and B.K. performed the measurements. A.M.R.V.L.M., T.C.T., H.Y., Y.X., A.D.C., and H.Y.H. fabricated the samples. E.P. and J.R. performed the RRN modeling. E.P., B.K., J.R, A.D.C., B.F., and K.B. interpreted the results. E.P, B.K., and J.R. wrote the manuscript, with contributions from all co-authors.

## Competing interests

The authors declare no competing interests.
