## [Peer Review File · Nature Communications]

Editorial Note: This manuscript has been previously reviewed at another journal that is not operating a transparent peer review scheme. This document only contains reviewer comments and rebuttal letters for versions considered at Nature Communications . Mentions of the other journal have been redacted. Parts of this peer review file have been redacted as indicated to remove third-party material where no permission to publish could be obtained.

REVIEWER COMMENTS

Reviewer #2 (Remarks to the Author):

I have carefully checked the manuscript and the replies to reviewers prepared by Persky et al. I feel all technical issues are adequately addressed by the authors, especially after multiple rounds of reviews in [Redacted].

Additional, I personally feel that study is novel, unique and comprehensive. I agree with some comments from the previous rounds on the novelty, but I think this work is definitely novel enough to be recommended to publish in Nature Communications.

Reviewer #5 (Remarks to the Author):

I have reviewed the manuscript, "Non-universal current flow near the metal-insulator transition in an oxide interface" as well as the reports of previous reviewers. The manuscript reports current density measurements, measured by scanning SQUID microscopy, of LAO/STO channels at and around the metal-insulator transition (MIT). The narrative focuses on the lack of expected scaling near the transition, and this breakdown of scaling is attributed to finite-size effects combined with the large scale of typical ferroelastic domain structures. The devices in question (shown in the supplement) are about 50um wide, and show clear ferroelastic domains that extend along the width of the channels. The authors point out (I am paraphrasing) that 50um is still mesoscopic, i.e., has properties that are determined by the boundaries of the device in question. I think that this is not a well appreciated fact, and the local measurements do a great job of hammering this point. With that said, I think that the way the result is described, it makes this reader wonder why mm-scale structures were not investigated. Why can't one image an extremely large device and recover the expected scaling?

I am also not convinced that the authors have in fact reached the insulating phase. The sensitivity of the SQUID is great for imaging, but doesn't have the resolution of straightforward transport measurements. I would prefer to see Figure R1 placed in the supplement at least. But it does not show an insulating transition, and with a resolution of 1um with scanning SQUID we cannot be sure that there are no thin channels left that are conducting at a resolution that falls below the dynamic range of the SQUID.

My last major comment concerns the statement that appears in the abstract and again in the text: "Our results offer a generic platform to engineer electronic transitions on the nanoscale." This is intriguing but I see scant evidence of such a platform. How are these transitions to be engineered? There does not seem to be any method of control.

Overall, I think that the manuscript could be suitable for publication in Nature Communications. But my concerns about the present form of the manuscript would preclude publication in the present form.

Reviewer #6 (Remarks to the Author):

The paper by Persky et al reports scanning SQUID measurements of the local conductivity of LAO/STO 2DEGs. The authors explore the role of domain boundaries on the metal-insulator transition and, in the light of these data and of results from random resistance network simulations, conclude that an accurate picture of the metal-insulator transition in this system can only be obtained by performing measurements at different length scales, hence the non-universal nature of the transition in this system.

The paper is quite technical [Redacted]. However, it may be suitable for Nature Communications.

One concern I have is that the authors consistently refer to the insulating phase of LAO/STO but to me it is unclear what insulating phase they are talking about. Is it the phase with a negative resistance vs temperature slope sometimes found at large negative voltage (cf. Ref 17) or the very high resistance insulating phase found for sub 4 uc samples? This should be clarified. Related to that point, the authors should also give more details about their samples, and in particular indicate the LAO thickness they have used (in the main text).

Continuing on this issue, it would be useful to show that such a gate-induced metal-insulator transition is achievable in their samples at the macroscopic level.

Also, one may wonder if the regions that the author call "insulating" are truly insulating or if they just have a higher resistivity. The authors should comment on this.

The authors mention that for the data of Fig 2a, the current flows in a mesoscopic device. However, the shape, geometry or dimensions of these device are not shown or mentioned, which is very puzzling.

The histograms of Fig. 2c are said to show "large variations". To me they actually look pretty similar, so this should be rephrased, and maybe the data should be plotted in a way to highlight better their differences.

When describing their model the authors list all the parameters that do not depend on the gate. I suggest they also explicitly state those that do.

In summary, this paper contains good data, some interesting simulations and discussions and could deserve publication in Nature Comms but contains some ambiguous formulations at this stage.

Reviewer #2 (Remarks to the Author):

I have carefully checked the manuscript and the replies to reviewers prepared by Persky et al. I feel all technical issues are adequately addressed by the authors, especially after multiple rounds of reviews in [Redacted].

Additional, I personally feel that study is novel, unique and comprehensive. I agree with some comments from the previous rounds on the novelty, but I think this work is definitely novel enough to be recommended to publish in Nature Communications.

We thank the reviewer for his/her comments through the review process, and for the encouraging feedback on the current version of the manuscript.

Reviewer #5 (Remarks to the Author):

I have reviewed the manuscript, “Non-universal current flow near the metal-insulator transition in an oxide interface” as well as the reports of previous reviewers. The manuscript reports current density measurements, measured by scanning SQUID microscopy, of LAO/STO channels at and around the metal-insulator transition (MIT). The narrative focuses on the lack of expected scaling near the transition, and this breakdown of scaling is attributed to finite-size effects combined with the large scale of typical ferroelastic domain structures. The devices in question (shown in the supplement) are about 50um wide, and show clear ferroelastic domains that extend along the width of the channels. The authors point out (I am paraphrasing) that 50um is still mesoscopic, i.e., has properties that are determined by the boundaries of the device in question. I think that this is not a well appreciated fact, and the local measurements do a great job of hammering this point.

We thank the reviewer for his/her positive view of our work.

With that said, I think that the way the result is described, it makes this reader wonder why mm-scale structures were not investigated. Why can't one image an extremely large device and recover the expected scaling?

Unfortunately, several experimental limitations prevent imaging extremely large devices in our setup:

1. For *quantitative* analysis of magnetic flux data, it is essential to image the entire cross section of the device and the magnetic fields generated away from it. Our microscope has a maximal field of view of ~400 μm , limiting the largest devices to 200 μm .
2. We are sensitive to current density. Devices 20-50 μm wide result in an ideal SNR, allowing us to apply smaller currents to the sample. With much larger devices, a higher current density is required to achieve comparable SNRs, limiting our ability to investigate the high resistance regime close to the MIT.

I am also not convinced that the authors have in fact reached the insulating phase. The sensitivity of the SQUID is great for imaging, but doesn't have the resolution of straightforward transport measurements. I would prefer to see Figure R1 placed in the supplement at least. But it does not show an insulating transition,

Thank you for pointing out that the transition was not clearly presented. We clarified the discussion in the text, and added supplementary figure S2, to demonstrate the phase transition through transport.

The onset of the insulating phase is characterized by a sharp drop in conductivity over a narrow range of V_G . Further, within the insulating phase there is weak temperature dependence and non-linear IV characteristics. These features have been well investigated in the past (See Figure R2 for examples from Refs. 17,20,21). We added a brief discussion to the manuscript:

Compared to semiconductor based 2DEGs²⁵, the MIT in LAO/STO is anomalous. First, the insulating phase in LAO/STO has a weak temperature dependence^{17,20,21}, and a crossover between $dR/dT > 0$ and $dR/dT < 0$ is often difficult to observe. Instead, the MIT is identified through a sharp rise in resistivity over a narrow range of gate voltages. The insulating phase is often accompanied by onset of non-linear IV characteristics²⁰. We studied the transport properties of two LAO/STO samples, and observed non-linear IV curves and large drop in conductivity below a critical V_G [Supplementary information], confirming our samples have bulk characteristics of a gate tunable MIT.

[Redacted]

Figure R2. Transport signatures of the metal-insulator transition in LAO/STO. **a**, Resistivity vs. temperature for various carrier concentrations in p-type GaAs/AlGaAs 2DEG. In this system, the insulating phase shows significant temperature dependence, with resistivity increasing by ~ 1.5 orders of magnitude from 1 K to 0.05 K. **b**, resistivity vs. carrier density in a LAO/STO interface, showing very no temperature dependence between 4 K and 35 K. The MIT is characterized by a sharp drop in conductivity, at a threshold carrier density. **c**, I-V curves for the device in b at various V_G , showing the onset of non-linear I-V characteristic below the threshold voltage. **d**, A similar sharp drop in conductivity is observed at an ionic-liquid-gated LAO/STO sample. **e**, $R(T)$ curves of an LAO/STO device show a gate tunable SIT at low temperatures. The insulating phase (blue curves) has a weak temperature dependence, with resistance increasing by a factor of less than 2 between 400 mK and 50 mK. Panel a is adapted from Phys. Rev. Lett. 80, 1288 (1998). Panels b,c from Ref. 20, panel d from Ref. 21 and panel e from Ref. 17.

In our devices, we observe transport behavior consistent with Figure R2. We added a new supplementary figure S2, where we show transport data of both the two-point resistance as a function of gate voltage, and IV curves, for two samples C1 and C3 (corresponding to SQUID data presented in main Figure 2 and

Supplementary Figure S4). The resistance increased sharply below a threshold V_G , where we also observed nonlinear IV. These measurements demonstrate an MIT in the overall transport properties.

Figure S2. Transport properties of the MIT in samples C1 and C3. **a,b**, zero-bias two terminal resistance of samples C1 (a) and C3 (b), as a function of gate voltage. Both samples show a sharp resistance increase below a threshold gate voltage. **c,d**, IV curves of samples C1 (c) and C3 (d), obtained for various V_G . Non-linear IV characteristics onset below the threshold voltages for the resistance increase. These data indicate the onset of an insulating phase at low V_G .

But it does not show an insulating transition, and with a resolution of 1 μm with scanning SQUID we cannot be sure that there are no thin channels left that are conducting at a resolution that falls below the dynamic range of the SQUID.

We thank the reviewer for bringing this up. At the percolation threshold, only half of the bonds are insulating. In the insulating phase, a substantial part of the sample may be conducting, but the sample still behaves as insulating since its two edges are not connected by a conducting path. Scanning SQUID provides the current density images, rather than the conductivity. We therefore explore the structure of the conducting paths that connect the two edges of the sample. This backbone only contains a small portion of the conducting nodes of the system. Areas of zero current density can also be, for example, metallic regions that are disconnected from the backbone and therefore do not carry current (also see our response to reviewer #6 and Figure R3).

The global properties of the system (namely, the resistivity) is controlled by the highest current carrying paths. So the possible existence of such paths does not prohibit an insulating phase, and does not change our conclusions.

We now explain this point in the main text:

Although the distorted current flow suggests a large inhomogeneity in the conductivity landscape, areas of low current density are not necessarily entirely insulating. There are several conductivity landscapes that can produce such current flow. One possibility is that these areas contain metallic regions disconnected from the current-carrying backbone. Another option is “dead end” conducting paths. Such paths originate from the backbone, but terminate at insulating regions. For such “dangling bonds”, we can estimate from the data the conductivity reduction at the insulating regions. The contrast of current densities on and off the backbone suggests that their conductivity is at most 1-5% of the conductivity of the backbone.

Regarding possible narrow channels, below the SQUID’s spatial resolution: even though the resolution is $\sim 1 \mu\text{m}$, we are able to detect narrower conducting channels, if they are isolated. For example, we previously imaged current flow $\sim 20 \text{ nm}$ wide carbon nanotubes [Nano Lett. 16, 4, 2152–2158 (2016)]. Since the current density we see outside the main backbone is low, it is more likely that metallic nodes in this area are connected to the backbone only through a very high resistance, otherwise the backbone would split into several channels. On the other hand, it is likely that the backbone we imaged has a finer structure that cannot be captured with our spatial resolution. This could lead to an extended low current tail of the histograms (Figure 2b,c). We clarified this in the new version:

We note that the current carrying backbone we observe is resolution limited, and may contain a finer structure. In this case, the maximal current density could be larger than the value extracted from the data, thus extending the low current tail of the histograms.

My last major comment concerns the statement that appears in the abstract and again in the text: “Our results offer a generic platform to engineer electronic transitions on the nanoscale.” This is intriguing but I see scant evidence of such a platform. How are these transitions to be engineered? There does not seem to be any method of control.

We meant that, given the dramatic effect domain structures have on the critical behavior, if the domain pattern could be controlled, the critical behavior could be engineered on the nanoscale. In ferroelectrics and ferromagnets, there is ample work on controlling domain structures [Rev. Mod. Phys. 84, 119–156 (2012)], and we expect some of the lessons learned there can be used in conducting interfaces. We are currently working on control methods for LAO/STO, and we hope our work inspires further studies in this direction.

We removed this sentence from the abstract, and added the following text to the discussion, to emphasize this is a future research direction emerging from our results.

This type of engineering could be a powerful tool for both fundamental and applied research. To achieve this, further work is required to improve the control over formation of domain patterns. Control over domain patterns (domain engineering) already has promising applications with ferroelectrics and ferromagnets⁴⁶, and techniques established for ferroics could be applied to heterostructures.

Overall, I think that the manuscript could be suitable for publication in Nature Communications. But my concerns about the present form of the manuscript would preclude publication in the present form.

We thank the reviewer for his/her constructive comments, and hope they find the revised version suitable for publication.

Reviewer #6 (Remarks to the Author):

The paper by Persky et al reports scanning SQUID measurements of the local conductivity of LAO/STO 2DEGs. The authors explore the role of domain boundaries on the metal-insulator transition and, in the light of these data and of results from random resistance network simulations, conclude that an accurate picture of the metal-insulator transition in this system can only be obtained by performing measurements at different length scales, hence the non-universal nature of the transition in this system.

The paper is quite technical [Redacted]. However, it may be suitable for Nature Communications.

We thank the reviewer for his/her time and helpful comments.

One concern I have is that the authors consistently refer to the insulating phase of LAO/STO but to me it is unclear what insulating phase they are talking about. Is it the phase with a negative resistance vs temperature slope sometimes found at large negative voltage (cf. Ref 17) or the very high resistance insulating phase found for sub 4 uc samples? This should be clarified.

We thank the reviewer for bringing up this ambiguity. Indeed, we refer to the gate tunable metal insulator-transition. We now explicitly state this in the manuscript:

We studied metallic interfaces, with LAO thicknesses of 10 u.c. and 12 u.c. In such samples, a metal insulator transition can be achieved by tuning the metallicity using electrostatic gating.

Related to that point, the authors should also give more details about their samples, and in particular indicate the LAO thickness they have used (in the main text).

We added Supplementary Table 1, which specifies the growth conditions of the samples included in this study. We studied four samples with LAO thicknesses of 10 and 12 unit cells.

Continuing on this issue, it would be useful to show that such a gate-induced metal-insulator transition is achievable in their samples at the macroscopic level.

We added the transport data for two samples, C1 and C3, to a new Supplementary Figure S2 (also pasted above). The data show (1) a sharp increase in resistance below a threshold V_G and (2) emergence of non-linear IV curves at the same threshold voltage. These are signatures of an MIT in LAO/STO (please see the response to reviewer #5, and Figure R2 for examples from previous reports, Refs. 17,20,21). We also added a brief discussion to the manuscript:

Compared to semiconductor based 2DEGs²⁵, the MIT in LAO/STO is anomalous. First, the insulating phase in LAO/STO has a weak temperature dependence^{17,20,21}, and a crossover between $dR/dT > 0$ and $dR/dT < 0$ is often difficult to observe. Instead, the MIT is identified through a sharp rise in resistivity over a narrow range of gate voltages. The insulating phase is often accompanied by onset of non-linear IV characteristics²⁰. We studied the transport properties of two LAO/STO samples, and observed non-linear IV curves and large drop in conductivity below a critical V_G [Supplementary information], confirming our samples have bulk characteristics of a gate tunable MIT.

Also, one may wonder if the regions that the author call “insulating” are truly insulating or if they just have a higher resistivity. The authors should comment on this.

This is a good point. The SQUID maps current flow, not conductivity. There are several possible conductivity landscapes, which can account for an observed current density map. Thus, we cannot determine whether areas that do not carry current in a particular V_G are insulating or have a higher resistivity than the neighboring regions. To further complicate things, near the percolation threshold, areas with zero current density could be metallic clusters which are not connected to the main current-carrying backbone (e.g. “dangling bonds”). The different types of clusters have been investigated before in the context of percolation: Figure R3 shows a cluster analysis of a square lattice at the critical point, demonstrating how large metallic clusters can be cast out of the backbone. Such areas will not show in a SQUID investigation, even though they are conducting, because they cannot carry current. We hope our work inspires further experiments with complementary local tools, to map the conductivity landscape.

[Redacted]

Figure R3. Cluster analysis at the percolation threshold. The metallic clusters of a 510 by 510 square lattice are assigned different colors. Specifically, the white cluster is the percolating backbone, which connects the bottom and top edges. Green clusters are “dangling bonds” – metallic clusters connected to the backbone at a single site. Even though these metallic clusters take up a substantial portion of the network, they cannot carry current. Figure from [Bunde, Armin, Havlin, eds. Fractals and disordered systems. Springer Science & Business Media, 2012].

That said, we can place rough constraints on the resistivity of the “insulating” areas, determined from the contrast between current density on and off the backbone. This estimation depends on what we assume is the conductivity landscape. For example, in an overall homogeneous current density with a local reduction (like Figure 1a), it is natural to associate the reduced current flow to an area with reduced conductivity (possibly a defect in the sample). A modulation like in Figure 1a can be explained by a conductivity roughly 50% lower than the surrounding. When the current flow is highly inhomogeneous (Figure 2a) it is more likely that the low current density is due to small insulating regions that “cut off” percolating paths from the backbone. In this case the conductivity of the insulating parts could be at most 1-5% of the conductivity of the backbone. A percolation transition does not require “insulating” regions to have strictly zero conductivity; a finite conductivity smaller than that of the metallic nodes is allowed [Phys. Stat. Sol. (b) 76, 475 (1976)].

We made several changes to the text to clarify these points:

1. We modified phrases such as “insulating patches” to “areas of reduced current density”.
2. We explained that the conductivity landscape cannot be determined directly from the SQUID data, and discussed the possible scenarios for the current flow observed in Figure 2.

Although the distorted current flow suggests a large inhomogeneity in the conductivity landscape, areas of low current density are not necessarily entirely insulating. There are several conductivity landscapes that can produce such current flow. One possibility is that these areas contain metallic regions disconnected from the current-carrying backbone. Another option is “dead end” conducting paths. Such paths originate from the backbone, but terminate at insulating regions.

3. We added the above estimations of changes to conductivity:

3.1. For figure 1:

In a mono-domain, on the metallic side, the current density was overall homogeneous, except for one local reduction (Figure 1a). In the metallic phase, such local region with reduced current density could reflect a local reduction in the conductivity (roughly 50% of the neighboring area), possibly due to defects. ~~due to a patch of reduced conductivity (Figure 1a).~~

3.2. For figure 2:

For such “dangling bonds”, we can estimate from the data the conductivity reduction at the insulating regions. The contrast of current densities on and off the backbone suggests that their conductivity is at most 1-5% of the conductivity of the backbone.

The authors mention that for the data of Fig 2a, the current flows in a mesoscopic device. However, the shape, geometry or dimensions of these device are not shown or mentioned, which is very puzzling.

We apologize for the confusion. The device imaged does not have an ordinary Hall bar geometry, but its width changes from 40 μm to 60 μm within the SQUID’s field of view. We clarified the caption of figure 2 to explicitly state that the solid white lines represent the edges of the device, and we added an optical microscope image of the device, as supplementary figure S3 (also pasted below).

Figure S3. Optical microscope image of sample C1. The bright (dark) areas are crystalline (amorphous) LAO. The interface below the crystalline layer is conducting. The dashed lines indicate the edges of the conducting pattern, the red rectangle indicates the scan area of Figure 2 in the main text. Within the scan area, the device width changes from 40 μm to 60 μm . Scale bar, 50 μm .

The histograms of Fig. 2c are said to show “large variations”. To me they actually look pretty similar, so this should be rephrased, and maybe the data should be plotted in a way to highlight better their differences.

We clarified that the variations refer to the width of the distributions, when comparing different mono-domains to the overall distribution. For example, the overall distribution had a much wider low-current tail. We replaced the phrase “large variations” with a more accurate description of the data:

At $V_G = -66$ V, histograms of different mono-domains vary ~~show large variations~~ with respect to the histogram of the overall image (Figure 2c).⁷ Particularly, the overall distribution contains a low-current tail extending one decade more than that of mono-domain 1. ~~particularly in the width of the distribution (Figure 2c).~~

When describing their model the authors list all the parameters that do not depend on the gate. I suggest they also explicitly state those that do.

The only parameter that depends on the gate is p , the fraction of metallic bonds. p is a parameterization of V_G , with lower p corresponding to more negative V_G . We now state this explicitly:

In this model, the metallic fraction, p , is a parameterization of V_G . Lower p corresponds to more negative V_G . [...] Thus, other than p , the model parameters do not depend on the gate.

In summary, this paper contains good data, some interesting simulations and discussions and could deserve publication in Nature Comms but contains some ambiguous formulations at this stage.

We thank the reviewer for his/her comments. We hope the revisions clarify the ambiguities, and that he/she will find the revised manuscript suitable for publication.

REVIEWERS' COMMENTS

Reviewer #5 (Remarks to the Author):

All of my concerns have been addressed in the last round of revisions. I recommend publication.

Reviewer #6 (Remarks to the Author):

I can now recommend publication in NComms.